# The Redesigning of Tires and the Recycling Process to Maintain an Efficient Circular Economy

**Dan Dobrotă** [1,*]**, Gabriela Dobrotă** [2]**, Tiberiu Dobrescu** [3] **and Cristina Mohora** [3]

[1] Department of Industrial Engineering and Management, Lucian Blaga Univesity of Sibiu,
550024 Sibiu, Romania

[2] Department of Finance and Accounting, Constantin Brâncuși Univesity of Targu Jiu,
210141 Targu Jiu, Romania; gabi.dobrota70@gmail.com

[3] Department Robots and Production Systems, Politehnica University of Bucharest,
060042 Bucharest, Romania; tiberiu.dobrescu@upb.ro (T.D.); cristina.mohora@upb.ro (C.M.)

[*] Correspondence: dan.dobrota@ulbsibiu.ro

**Abstract:** The redesigning of tires is addressed from two points of view, namely the structure of the materials and the constructive shape of these products. The necessity for research is justified by the fact that even during the product design stage, there may be situations that can irreversibly affect the separation of product components (rubber and insertion), and because it is strictly necessary to achieve the reuse and recycling of waste components. The proposed redesign refers to the inscription of the types of materials that are present in each area on the lateral surface of the tire. Thus, the new redesign has positive effects both economically and socially. To highlight these, a cost-benefit analysis (CBA) has been applied and the net present value (NPV) as well as the internal rate of return (IRR) have been determined for the classic scenario and for the two proposed scenarios. Testing the financial sustainability of the proposed solution was done through sensitivity analysis. An analysis of the new tire design from the point of view of the circular economy was also presented. The results obtained have highlighted the effectiveness of the proposed solution from a technical, economic, social, and protection of the environment point of view.

**Keywords:** tire redesign; circular economy; cost-benefit analysis; energy consumption; environmental pollution; sustainability

## 1. Introduction

Waste recycling has become an extremely important issue in regards to registering a larger quantity of waste due to its negative impacts on the environment. At present, waste in the form of used tires has become a real threat due to environmental pollution (satellite images of the Kuwait desert with black spots are relevant, which shows the world's largest landfill, containing over 7 million used tires) but also due to their effects on the health of the population (the incineration of rubber waste causes the release of large amounts of gas into the atmosphere) [1]. Also, the necessity of obtaining a growing volume of tire production, corresponding to the growing number of motor vehicles, requires the widespread use of virgin raw material (natural rubber) with a negative influence on the available natural resources. In this context, the application of technical solutions to increase the recycling of tires is an essential objective of the circular economy and, at the same time, a necessity [2].

In recent years, specific objectives in the field of environmental protection, namely minimizing waste, increasing recycling rates, or increasing product lifetimes, have become fundamental components of sustainable development strategies. However, it is obvious that their simple enunciation, without transposition into reality, is not enough. To achieve these goals, it is necessary to develop so-called green

engineering, which should focus on designing and delivering products using sustainable methods based on science and technology. Thus, the principles of green engineering should be taken into account, with the aim of designing new materials, processes, and products which provide optimum conditions for human health and the environment. Under these circumstances, designers should find solutions so that all materials and technologies used are environmentally friendly [3]. When designing products, the entire life cycle of the product should be considered in the sense that the product can be optimal from the point of view of environmental protection throughout its entire life cycle. However, the product may be obtained from environmentally hazardous materials, in which case the impact on it will be shifted to another part of the life cycle of the product. Thus, designers should also take into account the materials and energy inputs from the product obtaining system [2].

In the case where only the technology of obtaining the product is taken into account, there may be a potential design failure that causes significant risks to human health and natural systems. Engineers can use different principles for product design, representing guidelines used to make designs or models for products, processes, or systems to ensure the conditions and circumstances are necessary for sustainable development. In the first stage, when designing each product, designers need to assess the nature of the materials and energies used to ensure that an important milestone is reached to achieve a sustainable product. In a case where the selected materials and energies are not sustainable, their negative effects can be totally or partially canceled by optimized manufacturing technologies or the way the product is used [4].

Particularly important from the point of view of achieving a sustainable product is the fact that it is more beneficial to prevent unsustainable product achievements than to apply a treatment that makes the product a recyclable one. In many cases, products are not sustainable due to the fact that the processes and technologies applied are not efficient for the proper transformation of the materials and energies used. Under these conditions, products are obtained which later generate time, effort, and energy consumption waste [5].

The generation of certain quantities of waste is due to the lack of technology based on the design of the products, which allows a reduction of their waste quantities, but also allows a superior valorization of the resulting waste. Applying strategies in the field of design by using elementary design elements makes products, processes, and systems capable of being designed to prevent the generation of waste or the production of waste that can be valorized under optimum conditions [6,7].

An important issue that should be taken into consideration when designing products is the possibility of more easily separating the materials of which they are made. This category of products also includes the tires that have a rubber matrix, a metal insert, and a textile insert. The separation of tire components involves large amounts of energy and materials, but also quite complex technological processes [3]. Thus, certain known separation methods involve the use of large quantities of hazardous solvents, while other methods are based on high energy consumption in the form of heat or pressure. Under these conditions, it is advisable to design products that allow either the separation of components or the substantial reduction of energy consumption. A tire design strategy can be adopted to enable products to be produced using components with desirable properties but to allow easy separation during the recovery process. Such an approach can lead to a substantial reduction of material and energy consumption, but also to an increase in the efficiency of the waste recovery process [5].

Any decision taken at the product design stage can irreversibly affect the separation process of product components (rubber and insertion) in order to achieve reuse and recycling of components. At present, a number of technical and economic limitations represent major obstacles to the separation of tire components and these lead to a reduced recovery, recycling, or reuse of these types of waste. Establishing some preliminary component separation technologies from the tire design stage can create conditions for reducing material and energy costs from the recycling process [2,8].

Circular economic models are based on production chains built on technical cycles that are systematically able to ensure the maintenance, reuse, reconditioning and recycling of products and materials. In order to achieve significant results capable of avoiding climate change, circular economic

models need to be applicable and widespread on an extended geographic scale and for industrial processes that are resource-intensive [9].

When designing tires, it is necessary to take into account Le Châtelier's principle of minimizing the amount of resources consumed to obtain a product with desired characteristics. Thus, in the case of tires, an optimization of the quantities of materials should be made in the sense that any such product should contain corresponding amounts of rubber or insertion. If the proportion of the two components is not optimal, products with inappropriate usage characteristics and very high material and energy consumption can be obtained. Thus, this type of product should be designed to meet end-user demand related to product novelty, but also quality and quantity [8]. An important issue to consider when designing tires is to minimize material diversity. Thus, in the rubber matrix there may be different chemical additives, inclusive heat stabilizers, plasticizers, dyes, and flame retardants. This diversity becomes a problem when it comes to recycling the product at the end of its life cycle. The large variety of materials can cause difficulties in separating and recycling components. Thus, solutions should be adopted to minimize the material diversity of products, but to ensure the functions of each product at the same time [10].

In the design phase of the manufacturing process of the product, integration of product functionality should be considered to avoid the addition of additives at a certain stage of manufacture that could cause problems in recycling or reusing the product. Choosing the type of rubber into which certain additives are introduced can have a positive effect on the environment if it can be recycled more easily. For example, to reduce tire waste, it is important for certain parts of its structure, which remain functional and valuable, to be recovered for reuse and/or reconfiguration. A solution in this respect is the technological process of tire resurfacing, by maintaining the basic structure over which a new layer of new rubber is deposited, replacing the used rubber [1].

This strategy encourages so-called modular design in the sense that the rubber used for a particular part of the tire has to be homogeneous and have the same properties. When designing tires, it must be taken into account that these should have in structure components that can be recovered and that would significantly reduce the revaluation problems that occur at the end of their life cycle. Also, it should be taken into account that making rubber products, using large amounts of natural rubber, requires repetitive extractive processes, and this can cause resource depletion and environmental degradation [3]. However, natural rubber may be used within certain limits, provided that these repetitive processes are not environmentally damaging and at the same time the periods of regeneration of such resources are respected. When recycling tires, certain components (rubber, inserts) may be used as alternative raw materials for other types of materials (e.g., building materials), preserving the value of the raw material, but also the durability of the products [11,12]. The significance of the circular tire economy could be backed up by up to date waste tires statistics because over the last 15 years, there has been an overall growth in tire recycling and ELT (end of life tires) recovery [1].

In order to ensure a performant circular economy in the case of tires, the following conditions must be met [1,4]:

- the usage of raw materials and additives that can be recycled;
- redesigning products so as to ensure a more homogeneous structure of materials;
- redesigning the shape of the products in order to ensure their extended life span, but also to clearly observe the parts of the product that show a certain homogeneity of the structure;
- the establishment of a phase of designing the structure of products so that those parts of the product that remain functional and valuable after the product has been disposed of are recovered for re-use and/or reconfiguration into a new product.

Taking into account all these aspects, the research carried out aimed to redesign the tires so that there can be improved their circularity. Thus, the stages were as follows:

- proposing a redesign of tires by inscribing on their side surface the types of materials that are in each area;

   −    analyzing the effects of the tire redesign in terms of the recycling process;

   −    proposing two scenarios of tire recycling technologies that are considering the new redesign;

   −    cost-benefit analysis of the proposed scenarios and the setting of the optimal scenario in terms of sustainability, circular economy, and reduction of environmental pollution.

As efficient recycling of used tires is not currently carried out, the main objective of the research is to identify a solution for re-designing the tires and the recycling process, so as to ensure a sustainable economy of these types of products. In the context of the presented aspects, the paper was further structured as follows: Section 2—Redesigning the tires to ensure their efficient revaluation; Section 3—Cost-benefit analysis—framework; Section 4—Results and discussions, followed by Section 5—Conclusions, and Bibliographic references.

## 2. Redesigning of Tires and the Recycling Process to Ensure their Efficient Valorification

Tires are made up of four main parts: the tread, which is designed for contact with the ground and to ensure proper friction; the carcass, which is the structural part of the tire on which the tread is vulcanized; the shoulder, which minimizes the effects of irregularities of the terrain and transfers the load due to braking and oversteering under acceleration; the heels, which are used to fit the casing to the rim. All tires share several ingredients, as shown in Table 1 and Figure 1, with the basic ingredients being rubber elastomers, metal insertion, and textile insertion.

**Table 1.** Average composition of a tire [1].

| Ingredient | Rubber Elatomers | Carbon Black | Metal | Textile | Zinc Oxide | Others |
|---|---|---|---|---|---|---|
| Passenger car | 48% | 21.4% | 15.6% | 5.5% | 1% | 8.5% |
| Truck | 45% | 22% | 23% | 3% | 2% | 5% |
| Off road | 47% | 22% | 12% | 10% | 2% | 7% |

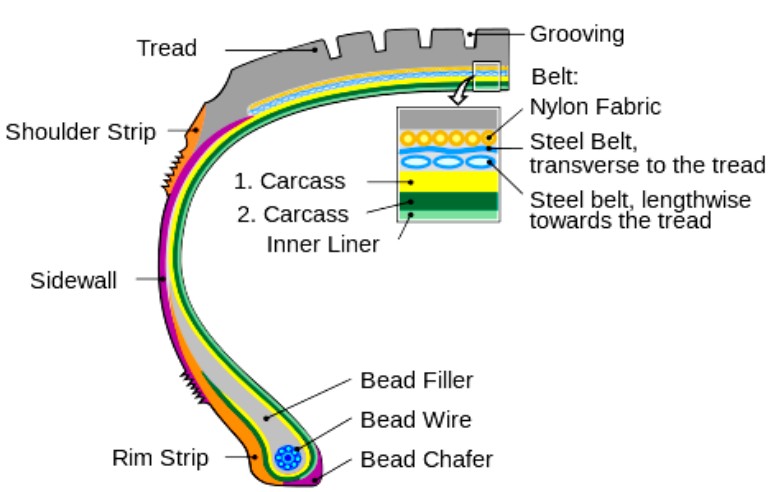

**Figure 1.** Tires structure [13].

For a superior tire revalution, it is important to design a technological process that allows the separation of the three main ingredients in their structure. Currently at the EU level, due to the problems of separating the three ingredients from the total waste collected, 58% of tires are recycled, 48% are used as a fuel source (75% in the cement industry and 25% in other industries) and 5% remain as residual waste [1].

From these data, it is observed that many tires are used as a fuel source in different fields. The EU policy on recycling non-recyclable and non-biodegradable waste is to limit its incineration. But in this category of waste also the used tires come in, so this policy is not transposed into practice for ELT (End-of-life tires). In this respect, new technologies need to be identified to allow a superior

revalution of ELT materials. In the case of conventional technology for tire waste recycling, five phases are necessary for the separation and recovery of tire ingredients, namely: backing-off, shredding, grinding, electromagnetic separation of metal components, and separation of textile components [7].

Thus, the first phase, represented by backing-off, involves the elimination of the tire part (hels), which has a large amount of metal insertion in the structure. For this stage, a special equipment is used that allows the capture of different types of tires and the cutting using knives of various geometries of this part of the tire. The second phase is represented by the shredding of the waste with a device containing a disk knife for cutting and another knife for shredding.

For shredding the waste, in the next phase, shredder equipment is used consisting of cylinders on which grinding knives are positioned to considerably reduce the size of the waste. After this phase is completed, an electromagnetic separation of the metallic waste is achieved which is a result of the shredding. For the cleaning of textile waste and obtaining cleaner rubber particles, a series of spray systems are used in the last phase. These systems are equipped with sieves of different sizes which allow the removal of both the textile component and the rubber particles. All of these phases that are routinely used to separate the components of the tires are shown in Figure 2.

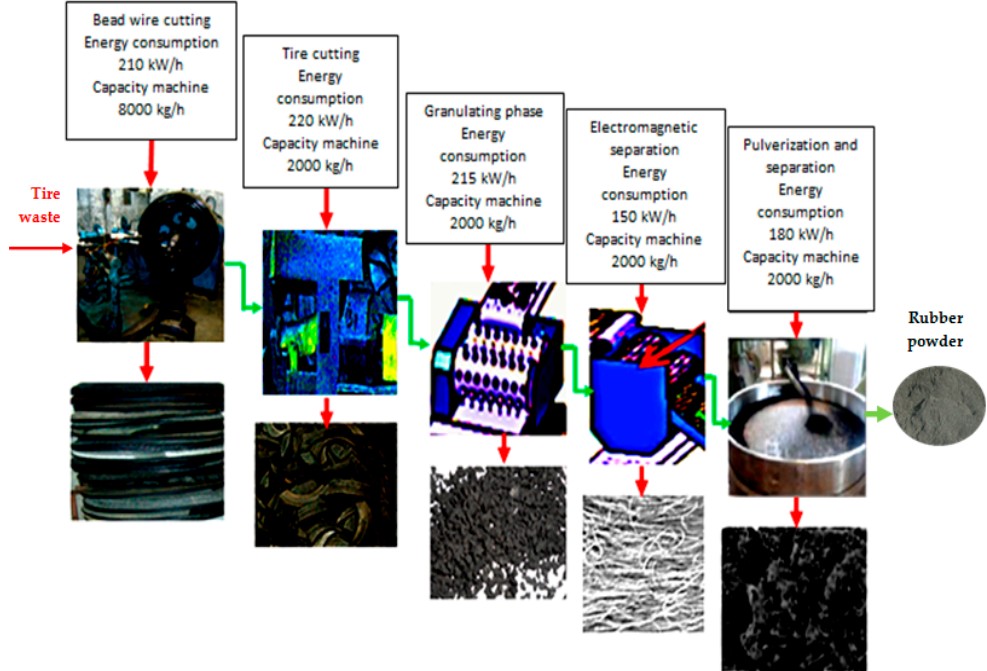

**Figure 2.** The disposal process waste tires in business-as-usual scenario.

Thus, in the case of the usual scenario, Figure 2, in the first phase, the tires are cut by separating the heels from the rest of the tire, Figure 3.

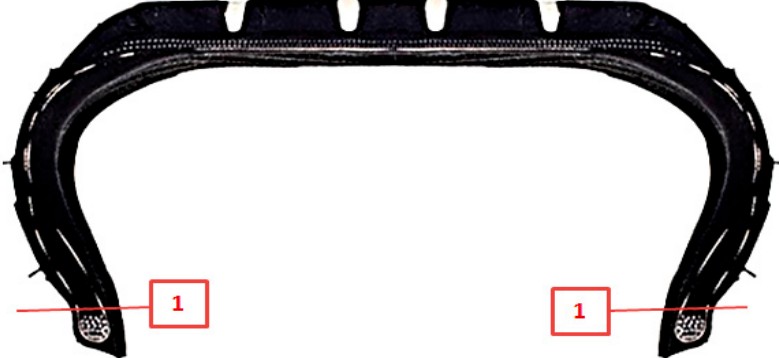

**Figure 3.** The usual scenario in which the tire is segmented in the first phase with heel separation from the rest of the tire.

This separation of the heels from the rest of the tire, in the case of the usual scenario, does not allow the creation of optimum tire recycling conditions. This is due to the fact that the remainder of the tire is cut and then shrunk without further separation of the tire parts. Applying only this separation, all three main types of ingredients (rubber, textile inserts, and metal inserts) are found in the remains of the tire.

Under these circumstances, there is a question of improving the usual scenario and proposing a new scenario in which it is necessary to carry out, in the first phase, a tire waste disposal in several parts. Thus, in the remaining tire pieces after the first phase, there will be no mixture of the three main ingredients in all the cuts. The new scenario assumes that in the first phase three cuts will be made after the three directions shown in Figure 4.

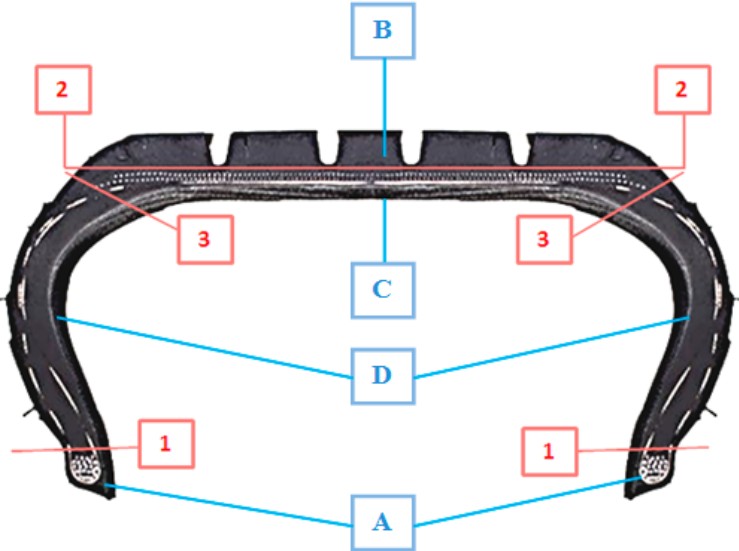

**Figure 4.** The new proposed scenario in which the tire is divided into several parts in three directions.

The cut in direction 1-1 results in a separation of the heels (A) from the rest of the tire, and in the heel part rubber, metal insertion and textile insertion is found. By cutting in direction 2-2, it is possible to remove piece (B) from the tire, leaving only rubber in the structure. This piece of the tire goes through a much simpler recycling cycle and allows the production of regenerated rubber with good properties. In the last stage, the tire is cut in direction 3-3, obtaining two distinct pieces: (C) and (D). In the tire piece (C), all three main ingredients in the tire structure (rubber, textile inserts, metal inserts) are found in the structure. As for the piece (D) it shows only rubber and textile insertion in

the structure. Due to this separation, the first phase of the usual scenario used for tire recycling is complicated to a certain extent, but the other phases of the tire recycling process are greatly simplified.

The analysis was conducted for a 185/65 R15 tire type with a normal wear and tear produced by Semperit, which is very commonly used in passenger cars. The analysis was performed for this type of tire, but the results can be used for other types of tires manufactured by various manufacturers. Depending on the type of tires and their manufacturer, the percentage of rubber or textile or metal inserts differs. In terms of the mass proportion of the four pieces cut from the tires, in the case of the proposed scenario, the four pieces have the following values: the piece A—10%; piece B—32%, piece C—20%; piece D—38%.

Thus, disposal technology through recycling for tire waste in the newly designed scenario is changing significantly, being able to obtain significant results in terms of energy consumption, reducing environmental pollution, and providing an optimal circular economy for tires.

Applying this new scenario has some limitations on tire design at the current time [1]. This also results from the fact that the environmental footprint of a product is determined from the design phase of the product. Lately, in the tire industry, the design of lighter, more fuel efficient, and more durable tires has been considered, using less resources for the same tire performance. However, this approach is not enough because designing the tires has not yet accounted for ensuring easy recycling. There are on the market tires with several inscriptions on the side surface, including the type of materials used (rubber, steel, nylon, etc.). However, this type of inscription is one that does not help much in the future tire recycling process. Thus, certain constructive modifications of the tires or their material should be carried out at the design stage. In this respect, it would be necessary to have either inscriptions or a series of ribbing on the outer surface of the tires which show the exact marking of the areas where the main tire ingredients are (rubber, textile inserts, and metal inserts). An example for the tires of the passenger cars is shown in Figure 5. Redesigning the tires, in accordance with the specified directions, can be done very easily, but the effects on the recycling process would be substantial and would help to provide the conditions to support their circularity.

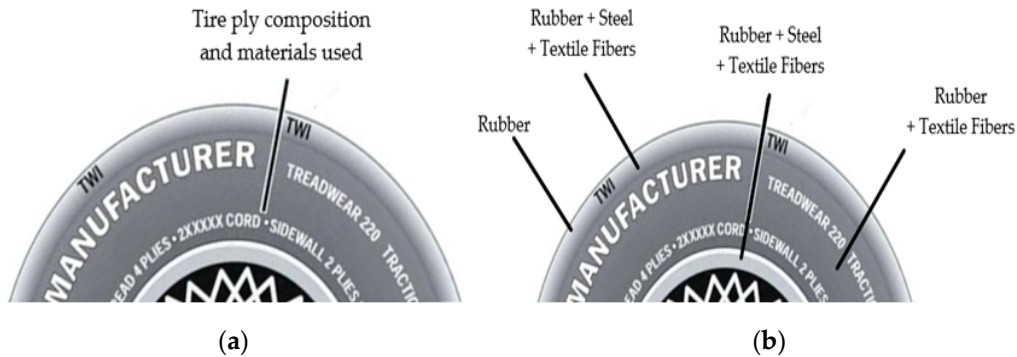

**Figure 5.** Tire inscriptions mode. (**a**)—the usual tire inscription mode; (**b**)—tire inscription mode used to apply the improved scenario.

In the circumstances where the redesigning of the tires is achieved, the necessary conditions for sustaining an efficient circular economy of the tire type products can be ensured and, in this way, a new scenario can be proposed. This new scenario is based on the fact that the tire has been redesigned and the three cutting directions of the tires shown in Figure 4 are exactly known. By applying the new scenario, a substantial change is made to the amount of rubber in the worn tire that has to go through the entire recycling technology stream. Thus, in phase 2 of waste shredding, only 30% of the tire weight is used, unlike the classic scenario when 100% of the tire mass is involved in this process. Also, only 68% of the weight of the tire is subjected to the separation process of the textile insertion, while 30% of the metal is subjected to insertion separation. All these changes to the usual scenario can contribute to reducing energy consumption, reducing environmental pollution, increasing the

percentage of used tires that are subjected to recycling, and providing the conditions needed for a circular economy, and sustainability of the tire waste recovery process. A situation regarding the stages of the tire waste recycling process for the improved scenario is presented in Figure 6.

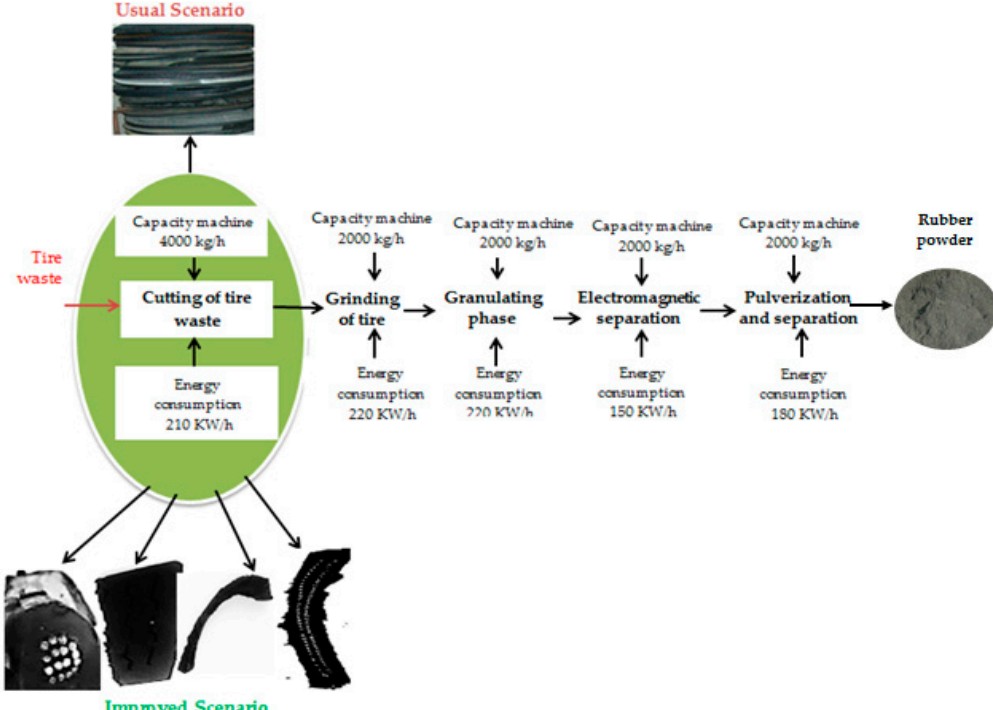

**Figure 6.** The disposal process waste tires in the improved scenario.

By applying the new tire recovery scenario in phase 1, the following waste categories result:

- 38% waste containing rubber and textile insertions (part D of the tire);
- 30% waste containing rubber, textile, and metal insertions (part C and A of the tire);
- 32% rubber waste (part B of the tire).

Considering this situation, some components resulted after Phase 1 do not fully participate in all phases following the technological process, respectively:

- the process of cutting the tires corresponding to phase 1 becomes one with a higher complexity, so that in this phase the tire cutting time increases and so does the other types of expenses; at the same time, due to the construction of the equipment used in this phase, minor additional investment is needed for the adoption of the new scenario, and these involve the adaptation of tools used for cutting and proper adjustment of the technological parameters. Also, the equipment used in this phase had a higher degree of loading in the usual scenario than the other equipment in the technological line; thus, with the adoption of the new scenario, a balance of the loading of equipment on the technological line is achieved;
- in phase 2 it is only necessary to cut only those components containing rubber, textile, and metal insertions and thus 30% of the weight of the tire is further processed at this stage;
- in phase 3 all the resulting waste products are shredded, but the productivity of processing increases because in the case of the usual scenario, the entire amount of waste resulting from the tire was the subjected to shredding by recirculation the materials three times (recirculation needed to remove the rubber from the insertions). In the case of the improved scenario, only 30% of the tire's material is subjected to shredding by three recirculation, and the remaining 70% of the resulting materials are shredded in one pass;

– at phase 4 when the electromagnetic separation of the metal insertion takes place, only 30% of the entire tire mass participates;

– only 68% of the entire tire mass contains textile insertions, so only this part of the resulting materials is subjected to processing in phase 5.

Applying the proposed waste revaluation scenario contributes to reducing the energy consumption required for re-use of tire waste. Thus, the distribution of the energy consumptions corresponding to each phase of the usual scenario and in the case of the improved scenario are presented in Table 2. The energy consumption calculations were based on the data presented in Figure 2, Figure 6, and the mode of participation of the tire waste within the 5 phases for the two scenarios.

**Table 2.** Energy consumption for one kg of tire waste in the case of the proposed scenario.

| Phase of the Technological Process | Energy Consumption in the Normal Scenario kW/kg Tire Waste | Energy Consumption in Case of Proposed Scenario kW/kg Tire Waste |
|:---:|:---:|:---:|
| 1 | 0.027 | 0.055 |
| 2 | 0.110 | 0.034 |
| 3 | 0.108 | 0.058 |
| 4 | 0.075 | 0.025 |
| 5 | 0.09 | 0.063 |
| **Total** | **0.410** | **0.235** |

From the data presented in Table 2, it is observed that by applying the proposed scenario the energy consumption for each kg of tire waste is reduced by 0.175 kW, so the energy consumption reduction rate is about 60%. Considering the production capacity of the technological equipments presented in Figure 2 and the fact that in about a year there are about 250 work days, it can be appreciated that on such a technological line 5000 tons of tires can be produced in one year.

Thus by applying the improved scenario, 1,408,000 kW of energy savings can be obtained for one year. If the average price of 0.20 Euro/kW is taken into account, then the financial gain resulting only from energy savings in the case of the proposed scenario is 281,600 EUR/year.

Applying the new ELT materials capitalization scenario may allow an increase in the share of ELT recycling; the proportion of recycled used tires in the EU has risen steadily from 16% in 1999 to 46% at present, while energy recovery was limited to 49% (ETRMA—European Tire & Rubber Manufacturers' Association). This figure takesa 40% reduction in energy consumption into account. Through the application of the new improved scenario, an increase in the share of ELT recycling can be obtained.

## 3. Cost-Benefit Analysis Framework

In the substantiation of the investment decision a very useful tool is the Cost-Benefit Analysis, as it ensures the identification of optimal variant of a project in terms of economic, environmental, and social effects as well as technological elements. The use of this method is considered to be opportune since the research made is aimed at getting positive environmental and social welfare effects in the context of the application of an efficient technology, through the prism of a benefit/cost ratio.

The principles of the Cost-Benefit Analysis (CBA) model first put forward by Jules Dupuit and developed by Alfred Marshal [14], underpinned the elaboration of the Kaldor-Hicks criterion, which is currently widely used to evaluate a project. Although the CBA is an analysis model used for governmental or local government projects or for private projects for which government or European funding is sought, in recent years it has been applied in the process of substantiating investment decisions at the level of private projects and, in particular, to those that have positive effects in economic, social, and environmental terms [15]. The utility of this tool derives from the fact that it is based on the comparative assessment of the costs of a project and its benefits over a projected period for at least two variants of the investment so as to identify both the economic efficiency and any amounts needed to ensure the project's realization and sustainability. In the Cost-Benefit Analysis, the term "costs" corresponds to negative cash flow and, also, negative impacts on society and nature, while

"benefits" are viewed in terms of positive effects on society, individuals, or the environment, as well as the positive flows (revenues) generated by a project. The monetary quantification of the effects of the project (economic, social, or environmental factors) and their comparison with the total expenditures generated allows us to identify the optimal solution for the realization of an investment.

In this context, it can be considered that the CBA can be used to identify the associated benefits and costs, determined by the application of the technical solution presented in the paper, so that the funding resources are established which are needed to ensure the sustainability of the project and the positive financial flows determined by recycling properly designed products. The presented model can be applied by the economic agents that carry out a recycling process of used tires, for which a corresponding redesign was achieved at the production stage. Thus, the use of this model offers the possibility of identifying problems that may arise before the project implementation and, implicitly, of making corrections to eliminate the identified deficiencies, as well as showing the benefits of the project's ability to ensure economic and financial sustainability of recycling used tires.

The main objective of this paper is the application of a technology that allows a superior revaluation of ELT materials, based on the redesign of tires. Thus, the cost-benefit analysis aims at analyzing the classical recycling process and the recycling process in the improved variant (both in the case of maintaining production capacity and increasing it). The expected results to be obtained when applying the improved version are:

- increasing the recycling rate of used tires;
- generating positive effects on individuals, communities, and the environment;
- reducing the cost of producing regenerated rubber;
- rational use of virgin raw materials and their replacement with material resulting from the recycling of used tires.

The construction and analysis of options for recycling used tires were based on the following steps: defining hypotheses, so as to ensure that financial projections for the operation period of the installation are achieved; identifying and assessing the costs of the current situation as well as the situation where is applied the recycling technology for used tires (for which the redesign process was previously carried out), corresponding to the technology used and the made materials; highlighting and monetary quantifying the benefits obtained (including positive flows generated by the sale of rubber powder, metallic waste, and textile fibers).

The research hypotheses taken into account in making the financial projections are:

- the use of the tire recycling facility for which the proposed redesign has been achieved leads to a reduction in the energy and transport costs and, implicitly, to reduced cost of producing regenerated rubber and an increase in labor productivity;
- the amount of used tires undergoing the recycling process increases;
- revenues earned on the sale of the resulting components increase, allowing a net positive value to be obtained.

Costs and benefits are quantified through inflows and outflows so that a comparison can be made between them to determine the annual surplus or deficit generated by the project implementation. Obviously, recording a surplus for the entire period would demonstrate the sustainability of the project. A summary of the benefits and costs of the proposed project is presented in Table 3.

**Table 3.** The financial flows generated by the project's costs and benefits.

| Financial Inflows (*B*) | Financial Outflows (*C*) |
|---|---|
| • Income from the sale of rubber powder<br>• Sums resulting from the recovery of metal waste<br>• Revenues from the sale of fibers | • The value of the investment<br>• Replacing cutting tools<br>• Operational costs<br>• Wages<br>• Transport costs<br>• Energy costs<br>• Fiber pressing costs<br>• Taxes on capital income and others |

Considering the importance of the financial, economic, and generated risk analysis, this research focused on the comparative analysis of the financial performance indicators for the three scenarios corresponding to the tire recycling process by applying the classical variant and the improved variant with and without increasing the amount of waste (Net Present Value—NPV and Internal Rate of Return—IRR), using updated values with a discount rate. Thus, the NVP was determined using the relation:

$$NPV = \sum_{i=1}^{n} \frac{F_i}{(1+a)^i} + \frac{V_r}{(1+a)^n} - V_I \tag{1}$$

where: $F_i$ represents the value of net cash flows; $V_r$—the residual value; $V_I$—the value of the investment; $a$—discount rate.

In the speciality literature, there are numerous studies on the choice of discount rates corresponding to the projects [16–20], but for the calculation of project-specific indicators and comparing them, the same 5% discount rate was used, in accordance with the recommendations of the European Commission guidelines [21] and with [22,23] the analysis interval being 10 years, limited by the lifetime of the facility used. In the substantiation phase of the investment decision, the discount rate hypothesis is extremely important, given that between the NPV and this rate, there is an inversely proportional relation in the sense that a rate increase causes a decrease in NPV and vice versa [24,25].

Likewise, net present values for estimated earnings and expenditures were established. Thus, for the determination of PVR (Present Value of Revenues) and PVC (Present Values of Costs), the following relations have been applied:

$$PVR = \sum_{i=1}^{n} \frac{R_i}{(1+a)^i} \text{ and } PVC = \sum_{i=1}^{n} \frac{C_i}{(1+a)^i} \tag{2}$$

where: $R_i$ represents the benefits (revenues); $C$—the costs; $a$—the discount rate.

The NPV determination also allowed the establishment of the relation between PVR and PVC so that the ability of the project to be able to provide benefits greater than the costs involved could be identified.

The internal rate of return has been established considering the equality in the following relation (practically, the IRR is the discount rate for which the NPV is null):

$$NPV = 0 \tag{3}$$

In order to test the stability of the proposed variant, a sensitivity analysis was also carried out. This required identification of the changes made to the indicators determined in the case of the forecasted changes in the values related to the elements considered in the cash flow dimensioning. Thus, an analysis of the risks involved in the application of the new project has been made from the point of view of the possibility of recording variations of NPV and IRR, as a result of changes in variables

during the analysis period. This type of analysis is widely used to identify the effects generated by changing a variable in a project on other variables under a certain prespecified probability level [26,27]. Thus, the decision maker may have a realistic reflection on the degree of sensitivity of the selected solution to the changes in the values initially considered over one or more parameters [28,29].

In fact, risk analysis is considered a fundamental part of project risk management as it provides information on the effects of project selection, facilitating the adoption of the investment decision and the integration of its sustainability [30]. Assessing the risks that may arise can be done using different methods. In this regard, numerous research focused on risk assessment through various techniques, including the hybrid MCDM technique [31,32], the Bayes model [31], the hybrid method SWARA-COPRAS [11], the fuzzy model based on the Elena guideline [24], or the Monte Carlo simulation [33].

The research carried out in this respect has imposed the following stages:

– identification of variables that can record important changes during the analysis period: revenue from the sale of recycled materials, costs, and updated cash-flows;
– the estimation of changes for each variable in the context of leaving the others unchanged;
– the recalculation of NPV and IRR;
– identification of variables that determine large amplitudes at the level of the two indicators, for variations of the values of variables ranging between ±1%;
– the comparison of the proposed variants for identifying the alternative characterized by the influenced indicators in the lesser proportion provided by the estimated changes of the considered variables.

## 4. Results and Discussions/Data and Results in Cost-Benefit Analysis

The cost benefit analysis was performed for three scenarios: the usual scenario and two proposed scenarios (PS I and PS II). In the case of the usual scenario and the PS I scenario, it was considered that a regular amount of waste was processed in one year, and in the case of the PS II scenario it was considered that a larger quantity of waste was processed by equipping the machines used for cutting with performance tools. The application of the PS I scenarios and PS II can be performed only under conditions where a redesign of the tires according to the ones shown in Figure 5 is performed.

In making the cost-benefit analysis, the following data (valid for the case of Romania in terms of personnel costs) were considered for the proposed scenarios:

– the reference period is 10 years, corresponding to the lifetime of the facility used;
– the investment amount is 400,000 Euros;
– gross average monthly salary is 1000 Euros/employee;
– the amount of waste processed annually is 5000 tons for the PS I scenario and 5700 tons for the PS II scenario;
– fiber removal costs 10 Euros/ton;
– proceeds from the sale of fibers are 80 Euros/ton;
– the amount received from the recovery of metal waste is 0.4 Euros/kg;
– the discount rate used is 5%.

Data on the materials obtained from the recycling of tires both in the classic version as well as in the case of the two proposed scenarios are shown in Table 4.

**Table 4.** Data of products made in the usual scenario and proposed scenarios.

| Usual Scenario | | Proposal Scenario I | | Proposal Scenario II | |
|---|---|---|---|---|---|
| **Products** | **Quantities** | **Products** | **Quantities** | **Products** | **Quantities** |
| Powder | 3945 tons/year | Powder | 3945 tons/year | Powder | 4498 tons/year |
| Metallic waste | 780 tons/year | Metallic waste | 780 tons/year | Metallic waste | 889 tons/year |
| Fiber | 275 tons/year | Fiber | 275 tons/year | Fiber | 313 tons/year |

The conducted analysis started from the hypothesis of processing the same amount of tires, under the condition of using products for which the redesign (proposal scenario I) was applied, but to a larger quantity (proposal scenario II). In the case of scenario II, a larger quantity of waste to be processed was considered, but this was only made possible by equipping shredding machines with tools that have superior technical performance. In the case of the proposed scenario II, we attempted to ensure a degree of loading of all the equipment in the technological line, considering that for the application of scenario I, only the first equipment in the technological line has a high loading degree, and the following equipment has a lower loading degree. The benefits and costs of the three scenarios are presented in Table 5 for the usual scenario, Table 6 for the proposed scenario I, and Table 7 for proposed scenario II.

**Table 5.** Costs and income in the usual scenario.

| Costs | Value—Euro | Income | Value—Euro |
|---|---|---|---|
| The amount of investment | 400,000 | Income from the sale of rubber powder | 1,183,500/year |
| Replacing cutting tools | 4000/year | Sums resulting from the recovery of metal waste | 312,000/year |
| Operational costs | 4000/year | Proceeds from the sale of fibers | 22,000/year |
| Wages | 60,000/year | | |
| Transport costs | 20,000/year | | |
| Energy costs | 369,000/year | | |
| Fiber pressing costs | 2750/year | | |

**Table 6.** Costs and income in proposal scenario I (PSI).

| Costs | Value—Euro | Income | Value—Euro |
|---|---|---|---|
| The amount of the investment | 400,000 | Income from the sale of rubber powder | 1,183,500/year |
| Replacing cutting tools | 15,000/year | Sums resulting from the recovery of metal waste | 312,000/year |
| Operational costs | 4000/year | Proceeds from the sale of fibers | 22,000/year |
| Wages | 60,000/year | | |
| Transport costs | 15,000/year | | |
| Energy costs | 211,500/year | | |
| Fiber pressing costs | 2750/year | | |

**Table 7.** Costs and income in proposal scenario II (PSII).

| Costs | Value—Euro | Income | Value—Euro |
|---|---|---|---|
| The amount of the investment | 400,000 | Income from the sale of rubber powder | 1349/year |
| Replacing cutting tools | 20,000/year | Sums resulting from the recovery of metal waste | 355,600/year |
| Operational costs | 7410/year | Proceeds from the sale of fibers | 25,040/year |
| Wages | 60,000/year | | |
| Transport costs | 17,100/year | | |
| Energy costs | 241,110/year | | |
| Fiber pressing costs | 3130/year | | |

*4.1. Financial Analysis*

The financial analysis for the three scenarios was achieved taking into account the following aspects: In the baseline scenario, energy costs are higher, according to the data presented in Table 2; transport costs are lower in the case of the proposed scenario due to the fiber compaction, the reduction of their volume and implicitly, the decrease in the number of transports carried out; the cost of replacing cutting tools is higher for scenarios I and II, as new types of tools are required by the process of cutting in the proposed variant (PSI) and the increase in the processed quantity (PSII). The discounted cash flow method (DCF) was used in the performed analysis, taking into account only the cash inflows and outflows (not other accounting items). Also, the analysis was performed for a period corresponding to the useful life of the equipment used. The investment and operational expenses, with the labor force involved but also those expenses represented by the replacement of the used components and the consumption of energy and transport, were taken into account. Cost elements and estimated revenues were used to determine important indicators that highlight the project's ability to generate benefits.

Revenues from the recovery of materials obtained from the recycling of tires have been established as follows:

- revenue from the sale of rubber powder = 3945 tons × 300 Euro/ton = 1,183,500 Euro/year;
- sums resulting from the recovery of metal waste = 780 tons/year × 0.4 Euro = 312,000 Euro/year;
- receipts from the sale of fibers = 275 tons/year × 80 Euro/ton = 22,000 Euro/year.

For Scenario II, estimated earnings were similarly established in line with the volume of materials resulting from the processing of a larger tire.

The data presented in Table 2 shows a significant reduction in energy costs due to improved processing technology from 0.410 kW/Kg of waste to 0.234 kW/Kg of waste and this is made possible by applying the redesign of tires.

The results obtained from the NPV, IRR, PVR and PVC indicators for the three scenarios are summarized in Table 8.

**Table 8.** The level of values recorded by the financial performance indicators for the three scenarios.

| Performance Indicators | Usual Scenario | Proposal Scenario I | Proposal Scenario II |
|---|---|---|---|
| Net present value (NPV) | 6,018,113.03 | 7,400,781.02 | 8,069,865.41 |
| Internal rate of return (IRR) | 193.15 | 222.58 | 256.4 |
| PVR | 11,131,846.12 | 11,131,846.12 | 12,690,964.78 |
| Present Values of Costs (PVC) | 3,550,067.63 | 2,380,224.79 | 2,666,623.94 |
| PVB/PVC | 3.13 | 4.67 | 4.75 |

From the data systematized in Table 8, it is observed that NPV increases by 16.05% for PSI and by 33.13% for PSII versus the usual scenario. If the IRR is considered, it is noted that it has higher values in both cases, which relate to the variant of waste processing for which the redesign process has been carried out. Consequently, both financial performance indicators demonstrate the need to apply the proposed technology. Thus, the NPV has positive values higher than the usual scenario and the IRR has higher values compared to the discount rate, which demonstrates the profitability of the proposed variants.

## 4.2. Risk Analysis

In order to identify the "critical" variables of the project (those that significantly influence its economic and financial performance, for which a variation of ±1% of the initial value determines a variation of more than 1% of the performance indicators), a sensitivity analysis is required.

From testing the stability of the proposed variant (in the case of the two scenarios), the systematized results were obtained in Tables 9 and 10.

**Table 9.** Financial performance indicators in the case of revenue change.

| Estimate Error | Proposal Scenario I | | | Proposal Scenario II | | |
|---|---|---|---|---|---|---|
| | **NPV** | **IRR** | **Present Value of Revenues (PVR)/PVC** | **NPV** | **IRR** | **PVR/PVC** |
| −1% | 7,307,273.51 | 220.05 | 4.63 | 7,963,261.31 | 253.11 | 4.71 |
| −0.5% | 7,354,027.30 | 221.49 | 4.65 | 8,016,563.36 | 254.76 | 4.73 |
| 0 | 7,400,781.02 | 223.44 | 4.67 | 8,069,865.41 | 256.40 | 4.75 |
| +0.5% | 7,447,534.80 | 224.88 | 4.70 | 8,123,167.46 | 258.04 | 4.78 |
| +1% | 7,494,288.52 | 226.32 | 4.72 | 8,176,469.51 | 259.69 | 4.80 |
| Min | 7,307,273.51 | 220.05 | 4.63 | 7,963,261.31 | 253.11 | 4.71 |
| Max | 7,494,288.52 | 226.32 | 4.72 | 8,176,469.51 | 259.69 | 4.80 |
| Difference | 187,015.01 | 5.82 | 0.09 | 213,208.21 | 6.58 | 0.09 |

**Table 10.** Influence of changes in expenditure on financial performance indicators.

| Estimate Error | Proposal Scenario I | | | Proposal Scenario II | | |
|---|---|---|---|---|---|---|
| | **NPV** | **IRR** | **PVR/PVC** | **NPV** | **IRR** | **PVR/PVC** |
| −1% | 7,420,774.91 | 224.05 | 4.72 | 8,092,265.05 | 257.09 | 4.80 |
| −0.5% | 7,410,777.96 | 223.74 | 4.70 | 8,081,065.23 | 256.75 | 4.78 |
| 0 | 7,400,781.02 | 223.44 | 4.67 | 8,069,865.41 | 256.40 | 4.75 |
| +0.5% | 7,390,784.07 | 223.13 | 4.65 | 8,058,665.59 | 256.06 | 4.73 |
| +1% | 7,380,787.13 | 222.82 | 4.63 | 8,047,465.77 | 255.71 | 4.71 |
| Min | 6,964,409.33 | 222.82 | 4.63 | 7,989,095.29 | 255.71 | 4.71 |
| Max | 7,004,721.42 | 224.05 | 4.72 | 8,035,050.42 | 257.09 | 4.80 |
| Difference | 40,312.09 | 1.23 | 0.09 | 45,955.13 | 1.38 | 0.09 |

From the data presented in Tables 9 and 10, it is observed that the financial performance indicators are not very sensitive to the estimated changes of the two variables R (revenues) and C (costs). Moreover, as shown in Figures 7–12, the stability of the proposed technology in terms of financial performance indicators is also demonstrated by the result obtained from the application of a combined scenario in which several variables change simultaneously and the influence on economic efficiency indicators is identified. Thus, for PSI, it is noticed that the main influence on the NPV is from income. At a variation of ±1% of revenues, the NPV has values ranging from 6,964,409.33–7,004,721.42, with a maximum amplitude of 187,015.01. From the Tornado diagram (Figure 7), it is noticeable that the NPV is less influenced than the other two variables considered.

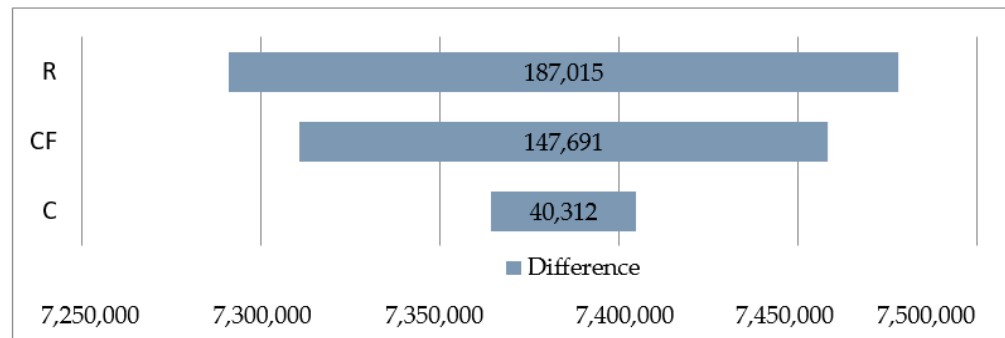

**Figure 7.** Sensitivity analysis tornado diagram—NPV PSI.

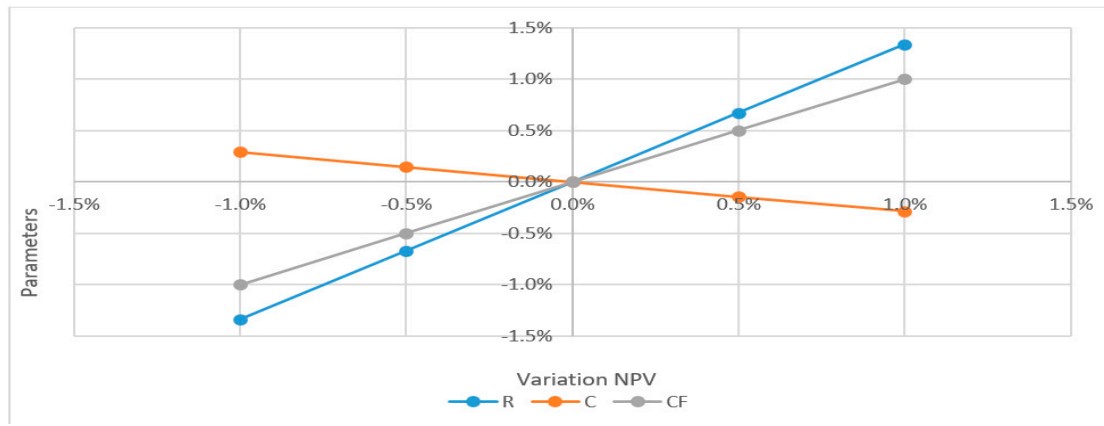

**Figure 8.** Sensitivity analysis spider chart—NPV PSI.

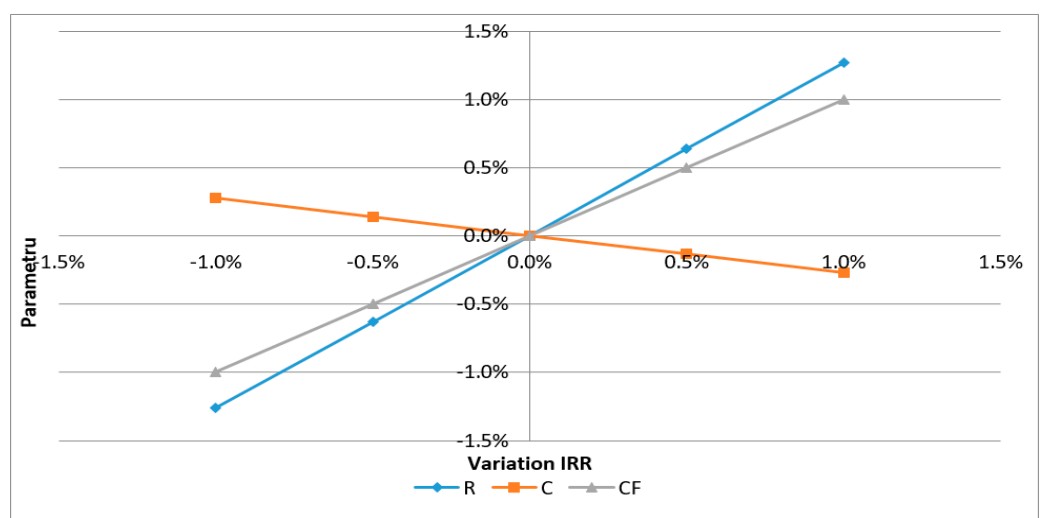

**Figure 9.** Sensitivity analysis spider—IRR PSI.

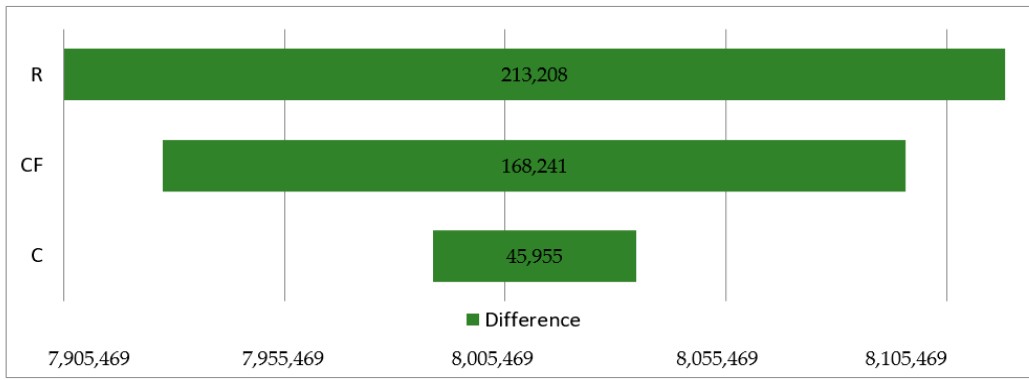

**Figure 10.** Sensitivity analysis tornado diagram—NPV PSII.

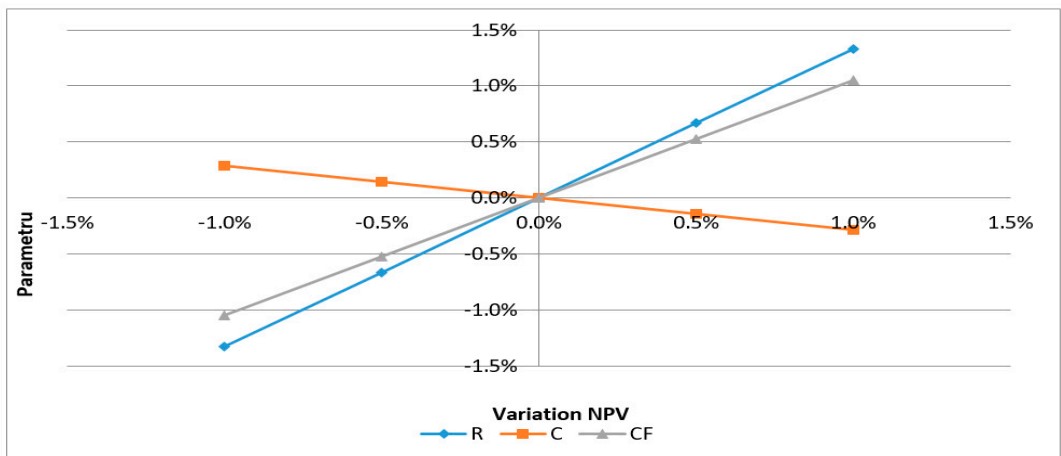

**Figure 11.** Sensitivity analysis spider chart—NPV PSII.

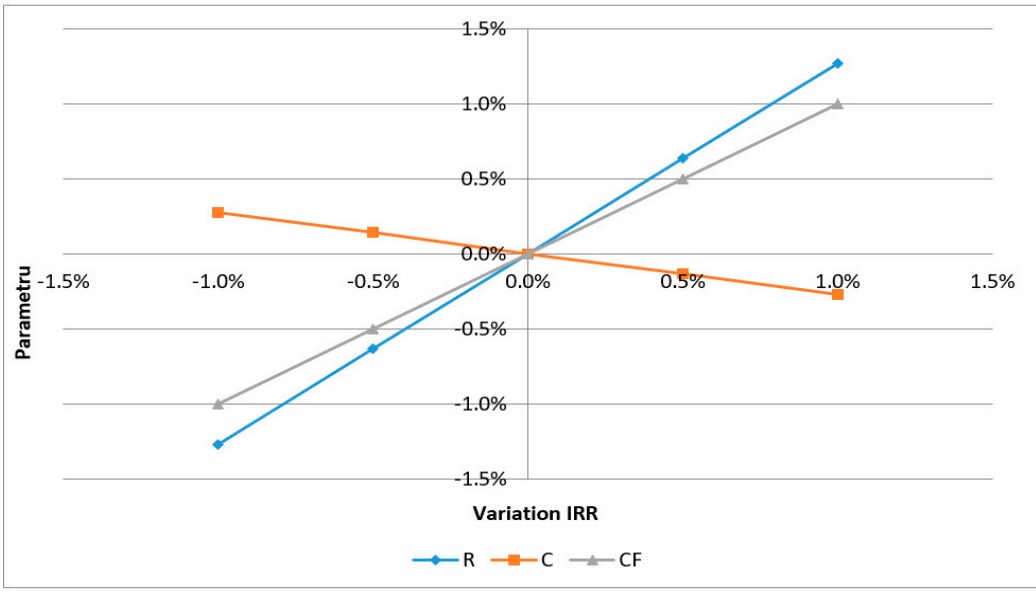

**Figure 12.** Sensitivity analysis spider chart—IRR PSII.

The percentage differences between the initial NPV and NPV influenced by the variance of the indicators in the analysis are reflected in Figure 8, making it obvious that the variables that should be considered are the incomes and the cash-flow.

The influence of the specified variables was also tested at the IRR level, as shown in Figure 9.

The analysis of the stability of the financial performance indicators in the case of PSII revealed that the revenues have the greatest influence on NPV and IRR, as shown in Figure 10.

The percentage change in revenue by ±1% in the PSII led to a change in NPV values by ±1.33%, while similar expenditure fluctuations generated a change of ±0.29% (Figure 11).

A similar situation was also observed in the case of IRR, which was influenced by income with a limit of 1.29%, and by expenditure in the ±0.28% interval, as shown in Figure 12.

The financial sustainability of the proposed scenarios is also demonstrated by the ratio which is registered between inflows and outflows. In the analysis, net financial flows are positive over the lifetime of the project implementation, even though the costs associated with the purchase of the facility are incurred in the first year.

Common practice recommends selecting a project that has a positive net present value, and if a comparison is made between two proposed scenarios, it is advisable to choose the one that provides the highest net present value. In line with this principle, the proposed scenario can be selected and achieved. In proposed version I, for every 1 Euro spent 4.67 Euros worth of benefits are obtained, while in variant II and the classic scenario, 4.5 Euros and 3.13 Euros of benefits would be obtained, respectively. As a result, PSI is recommended, since it shows the best results.

The redesign of tires and the adoption of their recycling technology by applying the PSII scenario create conditions for a superior treatment of these types of waste and the improvement of the conditions for ensuring the circular economy in the case of tires. Thus, due to the challenging conditions related to the recycling of tires, currently there is a poor circular economy for these types of products. Thus, Figure 13 shows that only 58% of used tires are currently recycled, and 42% of them are used as possible energy sources, of which 75% are used in the cement industry.

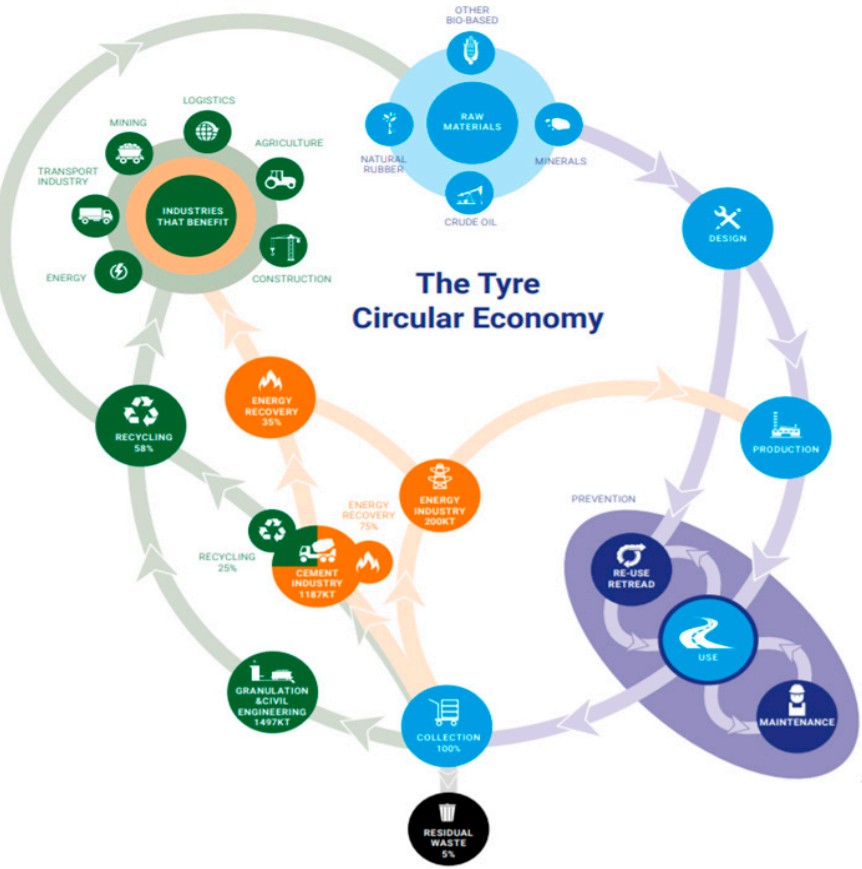

**Figure 13.** The circular economy of used tires [20].

By redesigning the tires and creating conditions for the PSII scenario application, conditions are created to provide the option for increasing the amount of recycled tires that can be recycled to a value of approximately 62% and reducing the amount of tires used as energy sources up to a percentage of 35%.

Thus, conditions are created where a smaller amount of tires can be used as an energy source in the cement industry and under these conditions, a reduction in global pollution can be achieved given that burning of tire waste in the cement industry leads to the release of large amounts of greenhouse gases, an amount which is estimated to be 5%–6% of all greenhouse gases generated by human activities [34]. Consumption of worn tires as a source of energy in the cement industry generates emissions into the air, including nitrogen oxides (NOx), carbon dioxide, chlorides, fluoride, sulfur dioxide, carbon monoxide, and lower amounts of organic compounds, and heavy metals [35]. All these emissions not only cause air quality degradation but also cause degradation of human health, while emissions have an impact on the local and global environment that causes global warming, ozone depletion, acid rain, loss of biodiversity, reduced crop productivity, and so on. Thus, it is possible to conclude that the redesigning of the tires ensures a high performance circular economy for this type of products, and also would lead to a reduction in environmental pollution.

## 5. Conclusions

This research had as a starting point the implementation of a process of redesigning the tires to facilitate their recycling process with positive effects in terms of reducing energy consumption in the waste recycling process and diminishing the negative effects of pollution of the environment. The proposed redesign of tires does not imply additional costs for their manufacture because it only involves inscription on the side surface of the tire of what types of materials are in each area. The application of the proposed solution leads to, in comparison with the currently used technologies, a decrease in the quantities of waste that are processed in the different phases of the recycling process. The positive effects of this are: diminution of energy consumption (there was a decrease of about 35% in the case of recycling tires with a new design), a reduction of environmental pollution (both through the reduction of the quantities of waste and of the level of the gases discharged into the atmosphere as a result of their incineration or used as fuel in various industries, respectively, mainly in the cement industry); bolstering the circular economy process, and ensuring the sustainability of the entire tire waste recovery process.

Also, the cost-benefit analysis highlighted the positive effects and financial performance of companies that opt for the implementation of the proposed technical solution, based on the redesign of the tires. Thus, the indicators determined for the two proposed scenarios (PSI and PSII) and taken into account in order to identify the financial sustainability compared to the classical variant, both for maintaining the quantity of processed waste and its increase (if the application of the proposed solution has the potential to process a larger amount of tires) have recorded higher and positive values compared to the usual scenario. The proposed scenario to be applied as a result of the NPV, IRR, and PVB/PVC ratio is PSII. Testing financial sustainability, carried out through sensitivity analysis, also demonstrated the efficiency of the proposed solution, given that the revenues and expenses generated by the proposed scenarios were considered as critical variables.

The proposed solution for redesigning the tires and, implicitly, the technological process of recycling them, demonstrates that an improvement of the circular economy for such products can be achieved. Also, by adopting these technical and economic solutions, a high sustainability level of the process of superior recovery of ELT with positive effects on environmental pollution can be obtained.

**Author Contributions:** D.D.—conceptualization and design experiment. G.D.—Cost-Benefit Analysis—framework; D.D., T.D. and C.M.—analyzed the data; D.D. writing—original draft preparation; D.D., D.G., T.D., and C.M. have read, corrected and approved the manuscript.

**Funding:** This research was funded by LUCIAN BLAGA UNIVERSITY OF SIBIU & Hasso Plattner Foundation research grants, grant number LBUS-IRG-2019-05.

**Acknowledgments:** Project financed by Lucian Blaga University of Sibiu & Hasso Plattner Foundation research grants LBUS-IRG-2019-05.

**Conflicts of Interest:** The authors declare no conflict of interest.

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
