# Peer review of "The Redesigning of Tires and the Recycling Process to Maintain an Efficient Circular Economy"

_sustainability, doi:10.3390/su11195204_

Round 1

Reviewer 1 Report

The paper presents a topic that refers to the redesign of the tires in order to achieve a higher capitalization of the tire waste under the conditions of minimum energy consumption.

From the analysis of the information presented in the paper I found the following:

- the paper, through its content, presents a series of information that can be useful to the scientific community;

- the introduction part is quite well structured, but other bibliographic sources could be considered;

- the differences between scenarios 2 and 3 should be clarified;

- unitary writing of the numerical values of the parameters from the cost-benefit analysis should be considered (for example table 8);

- the resolution of some figures should be improved;

- the part of conclusions could be completed so that own contributions can be highlighted.

Author Response

Dear reviewer,

We much appreciate your careful review. To improve the article, I have revised the article according to your suggestions. The changes and modifications in the manuscript have been highlighted.

Please note that the changes was made with red colour in text.

- in introductory part other bibliographic sources were considered;

- the differences between scenarios PSI and PSII were clarified;

- the numerical values were rewritten

- the resolution of some figures were improved;

- the part of conclusions were completed and own contributions were highlighted.

Finally, we are very thankful to you for taking your valuable time to help us with this paper. Your insightful and constructive advice and recommendations are deeply appreciated.

Reviewer 2 Report

please change the title due to two reasons: 1.it should state that the article presents the Romanian case (primarily because of the prices used in CBA). 2. your manuscript does not seem like it is about redesign of tires at all. It seems like redesign of recycling ELT (the way of cutting them) and only having certain preconditions when manufacturing new tires (side labels) that would enable proposed process your Introduction is more a Theoretical background. Please divide it in order to have both of those sections. Additionally, please add hypothesis, research question and it's justification in terms of the existing knowledge gap into the Introduction (you have those parts much later in the text) update your data in page 4: ETRMA annual report 2017 says 75% of ELT are used for energy recovery (86% in cement industry and 14% in energy industry) and 25% for material recycling in the same page you need a reference for 5 phases mentioned.In addition, please be specific and say that those 5 phases concern only material recovery, not energy recovery  for the presented way of cutting in page 6 please be specific and say that it is your proposal that you tested  it is not enough to say that the analysis was conducted on a tire 185/65 R15. By whom? Where and when? Only one tire? It's not quite relevant to base the whole study on a single tire. in the section Cost benefit analysis there are problems with punctuation. Please fix that You entitled that section CBA framework, therefore please organize it more transparently into Financial analysis, Economic analysis and Risk analysis. This shall require having some parts from Results and discussion moved here. you say costs in CBA refer to negative cash flows. That's true only for financial analysis. However, in economic analysis they include also negative impacts on society and nature. your text in page 11 says the discount rate has been chosen after European Comission guidelines, but your references (20,21) are not direct. Please consult "Guide to Cost Benefit Analysis of Investment projects" published by European Union in 2014, page 55  

Author Response

Dear reviewer,

We much appreciate your careful review. To improve the article, I have revised the article according to your suggestions. The changes and modifications in the manuscript have been highlighted.

We changed the title We specified in the introduction the objective of the research We updated the data on page 4 We made clarifications about the type of tires used For the 5 phases we added the bibliographic source and we made the necessary specifications The analysis was performed for this type of tire, but the results can be used for other types of tires manufactured by various manufacturers. Depending on the type of tyres and their manufacturer, the percentage of rubber or textile or metal inserts differs. We reviewed the punctuation in section Cost benefit analysis We have structured the section Cost benefit analysis We have corrected the specification regarding the negative cash flows We referred to Guide to Cost-Benefit Analysis of Investment Projects We changed the reference for figure 13 and corrected the data according to Annual Report 2017 of ETRma.

Please note that the changes was made with red colour in text.

Finally, we are very thankful to you for taking your valuable time to help us with this paper. Your insightful and constructive advice and recommendations are deeply appreciated.